# Enhancement of Convection and Molecular Transport into Film Stacked Structures by Introduction of Notch Shape for Micro-Immunoassay

**DOI:** 10.3390/mi15050613

**Published:** 2024-04-30

**Authors:** Daiki Arai, Satoshi Ogata, Tetsuhide Shimizu, Ming Yang

**Affiliations:** Department of Mechanical System Engineering, Faculty of System Design, Tokyo Metropolitan University, Tokyo 192-0397, Japan; daiki.arai@gmail.com (D.A.); ogata-satoshi@tmu.ac.jp (S.O.); simizu-tetuhide@tmu.ac.jp (T.S.)

**Keywords:** ELISA, microfluidics, 3D-stack, notched-shape, unsteady rotation

## Abstract

A 3D-stack microfluidic device that can be used in combination with 96-well plates for micro-immunoassay was developed by the authors. ELISA for detecting IgA by the 3D-stack can be performed in one-ninth of the time of the conventional method by using only 96-well plates. In this study, a notched-shape film was designed and utilized for the 3D-stack to promote circulation by enhancing and utilizing the axial flow and circumferential flow in order to further reduce the reaction time. A finite element analysis was performed to evaluate the axial flow and circumferential flow while using the 3D-stack in a well and design the optimal shape. The 3D-stack with the notched-shape film was fabricated and utilized for the binding rate test of the antibody and antigen and ELISA. As a result, by promoting circulation using 3D-stack with notched-shape film, the reaction time for each process of ELISA was reduced to 1 min, which is 1/60 for 96 wells at low concentrations.

## 1. Introduction

There are more opportunities to encounter unknown pathogens in modern society, and emerging infectious (EI) diseases are emerging almost every year [1]. The speed at which pathogens spread is increasing due to the speed and mass movement of people and goods, and there is a high possibility that they will spread over a wide area in a short period of time. For the prevention of the spread of these infectious diseases, it is necessary to prevent pathogens from entering from overseas via aircraft and ships, and early detection and isolation of infected people at airports is also important. In addition, there are cases where multiple patients occur sporadically on a small scale in the spread of infection, and it is important to follow up on the route of infection [2]. Therefore, the diagnosis of infectious diseases requires a speedy analysis that can instantly test for the presence or absence of infection in many people, and the analysis should have high sensitivity so that it can detect even when there are few pathogens in the early stages of infection. 

Immunoassay is a biochemical test commonly used in hospitals and laboratories. It has been used for many types of tests, from home pregnancy tests to AIDS tests to recent COVID-19 diagnoses [3,4,5]. Immunoassays are classified into multiple types according to label form, such as enzymes, radioisotopes, fluorescent dyes, chemiluminescent probes, etc. Enzyme-based labels, which are easy to molecularly design for the entire assay, are the most used, and they are especially used for ELISA (enzyme-linked immunosorbent assay). ELISA immobilizes a primary antibody on the surface of a polystyrene 96-well plate to detect a specific antigen in the sample. Although this method has high specificity and sensitivity and the ability to perform multiple assays at a time, antigen–antibody binding takes a long time because antigen diffusion is dependent on Brownian motion [6,7]. On the other hand, microfluidic devices are used to create microfluidic channels and reaction vessels using microfabrication technologies such as MEMS technology and apply them to biotechnology and chemical engineering. In the case of ELISA, the use of microfluidics is expected to shorten the analysis time and increase sensitivity by shortening the diffusion distance to the captured antibody and expanding the specific surface area of the reaction field. Various microfluidic devices, such as channel patterning [8], microbeads [9], centrifuge disks [10], and paper-based devices [11,12], have been developed for ELISA. Although a variety of microfluidic devices have been developed, most laboratories still use the conventional 96-well plate for these assays [13]. This is due to the high cost of device fabrication [14] and the need for peripheral equipment, such as pumps and detectors [13]. In medical and biological applications, disposable devices are desirable to avoid biological contamination and false-positive signals; therefore, low-cost and mass-producible microfluidic devices are required [15]. Even if the manufacturing cost of the device can be reduced, the conventional plate reader cannot be used, and a detector needs to be developed for quantification.

A 3D-stack microfluidic device was developed by the authors. Multi-annular films were stacked with a certain interval to form a 3D-stack structure which was produced in large quantities and at a low cost by a stamping process. By placing and rotating it into a well in the 96-well plate, the 3D-stack structure can work as a microfluidic device by circulating liquid within the well and creating liquid flow in the gaps between the stacked films due to the centrifugal force [16,17]. Compared to conventional micro-channels, the most important features of 3D-stack are that a micro-flow can be generated by rotating it, and the structure can be simply manufactured by a conventional stamping process; in addition, for the ELISA processes, an antibody can be immobilized on the surface of the 3D-stack films instead of the surface of the well. As a result, the reaction takes place on surface of the films, and the processes can be switched by moving the 3D-stack from one well to another well with different reagents; automation of the assay operation could be much simpler and easier than the conventional operation system. Figure 1 shows a schematic diagram of the 3D-stack. An analysis was performed by driving 16 3D-stacks simultaneously, using the motor rotation driving equipment. The ability to drive 16 3D-stacks simultaneously with one motor dramatically increases the utility of the 3D-stack method, and it is possible to proceed with a more parallel analysis by increasing the number of driving units for commercialization. In a previous study using the 3D-stack, Maeno et al. clarified the inhibitory factor by serum, detected CD163 in serum samples by covalently binding the complement antibody by EDC-NHS coupling, and showed a high correlation with the conventional ELISA in the detection of CD163 in the serum of dengue fever patients [17]. Since the molecules accumulate in the outer periphery of the container while the structure is rotated in the steady state, molecular transport between films becomes restricted and, thus, a bottleneck for reduction of measurement time. To reduce the process time, an operation by stopping and re-rotating the rotation of the annular film was performed to generate secondary flow to enhance the diffusion of molecules. It was shown that IgA can be measured in one-ninth of the time of the conventional method from the diffusion promotion [18]. It may be possible to promote convection, as well as diffusion, in order to further speed up the process. However, transportation of the molecules to the center of the 3D-stack is necessary in the case of the current annular shape. Since it is inhibited by the circulating flow due to rotation, the circulation condition in the container and between the films is poor, and how to form a circulating flow of biomolecules is still an issue.

There are two elements for this issue: rotation mechanism and geometry. Regarding the rotation mechanism, an increase in the rotation speed for the promotion of convection does not contribute to reflux in the axial direction and induces a vortex at the top of the 3D-stack, contributing to the exposure of the film surface and the generation of bubbles. Regarding the geometry, the combination of the annular film and rotating mechanism used in the previous study is similar to the Tesla pump. A Tesla pump is a type of turbomachine that uses a rotating impeller made of circular plates stacked at equal intervals and creates a flow from the pressure difference created by the rotation [19]. However, the current annular shape is not suitable for promoting circulation. To create an axial flow to promote convection, we focused on the shape of the reactor’s impeller. A reactor is a device that performs a chemical reaction, and an impeller is a rotating part that improves the mass transfer rate and uniformity of the fluid in the reactor [20]. The shape was devised to promote axial flow and improve mass transfer speed. In particular, the shape of a propeller, which has a space in the axial direction and forms a flow path in the axial direction, could be a shape suitable for forming axial flow in a stacked film structure.

In this study, we focused on geometry as a convection promotion of 3D-stack and attempted to design the geometry by adding a notch to the film shape, like a propeller, to promote an axial flow at the notch part and shorten the distance of the flow per unit time to circulation to the intervals of the film. Furthermore, by adding a notch, it was assumed that the circumferential flow from the notch would be used as an inflow between films, so it was expected to improve the circulation condition in the container and between films. Specifically, we focused on the axial flow caused by the addition of a propeller-like notch to the film to promote convection and the inflow due to circumferential flow at the notch part and attempted to elucidate the effect of convection promotion by the notch shape on biomolecule transport. We constructed a numerical analysis model and evaluated the effect of biomolecular transport by analyzing the circulating flow, e.g., the circumferential flow between films. Furthermore, we verified the changes in biomolecular transport due to convection promotion in ELISA, using a 3D-stack with a notch shape experimentally.

## 2. Materials and Methods

### 2.1. Design and Fabrication of 3D-Stack

For the enhancement of convection during the unstable rotation of the structure, a notched shape was proposed. Figure 2 shows the dimensions of the proposed 3D-stack with notches, consisting of five 100 μm thick PET films (Polyester Film Lumirror™ (Toray Industries, Inc., Tokyo, Japan) stacked in intervals of 20 μm to 100 μm and a fan-shaped notch applied to the disc film. In order to minimize the reduction of the reaction field area and to maintain mechanical strength, a fan shape that was cut in the normal direction was used. In the notch shape, a metal pin with a radius of 0.5 mm was penetrated to the center to stabilize the rotation in order to allow the circumferential flow from the notch to flow in. The dimensions of the wells were based on the drawings of the ELISA plate 96F S (MS-8496F, Sumitomo Bakelite Co., Ltd., Tokyo, Japan). The notch-shaped 3D-stack is referred to as 3D-stack (N), and the annular-shaped 3D-stack shown in Figure 1 is referred to as 3D-stack (C). The notch-shaped films with a diameter of 5 mm were punched from a PET sheet and then stacked at a regular interval after dowel processing for keeping the interval between the films. The 3D-stack structure was fabricated automatically by using micro-press forming equipment developed by the authors [21].

### 2.2. Mathematical Models and Numerical Analysis

To evaluate the formation of circumferential flow between films of the 3D-stack (N) and the molecular transport state, a finite element analysis was performed using the software COMSOL Multiphysics ver. 6.0 (COMSOL AB Inc., Stockholm, Sweden). Since the number of elements in a three-dimensional analytical model, including containers and structures is large, and the calculation time is enormous, we created a geometry only between films and applied the centrifugal force and Coriolis force as volumetric forces to verify the biomolecular transport state and the effect of the stacking spacing between films during rotation. A schematic diagram of the analysis model is shown in Figure 3.

#### 2.2.1. Flow in Film Gap

The flow is assumed to be laminar flow, and the incompressible Navier–Stokes equation and the continuous equation shown in the following equation are solved.
(1)$$\frac{\partial u}{\partial t}+\left(u\cdot \nabla \right)u=-\frac{1}{\rho }\nabla p+\upsilon ∆u+f$$
(2)div u=0
where *u* is the flow velocity, *p* is the pressure in the fluid, *ρ* is the mass density in the fluid, *ν* is the kinematic viscosity coefficient in the fluid, and *f* is the sum of the forces acting directly on the fluid from the outside. Since the flow is formed between films by the centrifugal force applied by the rotation, *F_w_*, and Coriolis force, *F_c_*, the flow due to rotation was defined from the following two equations [22,23,24].
(3)$$F_{c}=2\rho \omega u_{r}$$
(4)$$F_{w}=\rho r\omega^{2}$$
where *ω* is the angular acceleration, and *u_r_* is the radial velocity. The rotation speed is 2000 rpm, and the rotation is constant, with no time change.

#### 2.2.2. Bulk Analyte Transport

In order to model mass transport, it is necessary to describe not only flow but also changes in concentration in solution and consider flow and diffusion. Here, we used the advection–diffusion equation, Equation (5). In this analytical model, molecules are treated as bulk, like concentrations, not particles.
(5)$$\frac{\partial {\left[C\right]}_{bulk}}{\partial t}=\nabla \cdot \left(D\nabla {\left[C\right]}_{bulk}\right)-\nabla \cdot \left(u{\left[C\right]}_{bulk}\right)$$
where *C* is the concentration of the antigen on the bulk phase, *D* is the diffusion coefficient, the first term on the right side represents diffusion due to the concentration gradient, and the second term represents advection.

#### 2.2.3. Surface Binding Kinetics

An antibody is a protein that specifically binds to a foreign substance that has invaded the body, and the invading foreign substance is called an antigen. The binding reaction between the antigen and the antibody is called the antigen–antibody reaction, and ELISA uses this reaction to detect a specific molecule [25].
(6)$$\frac{\partial \left[C_{s}\right]}{\partial t}=D_{s}\nabla^{2}\left[C\right]+k_{on}\left(\left[C_{0}\right]-\left[C_{s}\right]\right)\left(\left[B_{0}\right]-\left[C_{s}\right]\right)-k_{off}\left[C_{s}\right]$$
where *B*_0_ is the concentration of the free binding sites at the surface, and *C_s_* is the concentration of the antigen–antibody complexes at the surface. *D_s_* is the surface diffusion constant, *k_on_* is the binding rate constant, and *k_off_* is the dissociation rate constant, respectively. The first term on the right side represents the diffusion on the surface, and the second and third terms represent the reversible reaction between the antigen and the antibody.

#### 2.2.4. Computational Environment and Analytical Model

In this study, the biomolecular transport analysis includes not only flow but also advection–diffusion equations and antigen–antibody reactions, and a time-dependent analysis was used. As boundary conditions, the wall surface was set to be non-slipping, and an open boundary was defined at the boundary between the container and the film, where fluid flows in and out between the films. The open boundary is the boundary where the fluid is in contact with a large amount of fluid, and the normal stress is 0 N/m^2^. In addition, in order to reduce the computational load, the symmetry boundary was defined, and the number of elements was reduced. Next, the inflow concentration was always constant at 0.066 nM with respect to the open boundary. This corresponds to 10 ng/mL of Human IgA used in the experiment. The numerical values of the parameters used in the analysis are summarized in Table 1. The surface density of the captured antibody and the diffusion coefficient of the antigen antibody were chosen in reference to the paper by Nygren et al. [26,27,28].

Furthermore, since the circumferential flow is a flow caused by inertia, it is expected that the circumferential flow will be improved by the change in Reynolds number due to the stacking interval. Therefore, a total of five geometries, namely 20, 40, 60, 80, and 100 μm, were created as lamination spacing. A mesh was created with hexahedral elements (Figure 4). To evaluate the influence of mesh size, the thickness (axial direction) of the mesh was changed to 10, 5, 2.5, 1.25, and 0.625 μm, and the concentration of antigen–antibody conjugates on the surface of the film was evaluated as a feature. As a result, the value stabilized at 2.5 μm or less. Therefore, the mesh size was set to 1.25 μm, and the mesh was created (Table 2). The PC for analysis was a Precision 7920 (DELL Inc., Round Rock, TX, USA), and the CPU was Intel^®^. Xeon^®^ Gold 5218R (20 cores, 2.10 GHz), 12 slots of 8 GB DDR4, 96 GB in total.

### 2.3. Materials and Methods for ELISA

The proteins, buffers, and substrates used in the ELISA were purchased from Bethel Laboratories (now Fortis Life Sciences, Waltham, MA, USA). The buffer solution and wash solution were prepared by diluting ELISA Coating Buffer (E107), ELISA Blocking Buffer (E104), and ELISA Wash Solution (E106) with distilled water. TMB One-Component Substrate (E102) was used as the substrate, and TMB Stop Solution (5150-0020) by SeraCare Life Sciences (Gaithersburg, MD, USA) was used for reaction stop. Complementary antibody binding to the surface of the structure was performed using an EDC-NHS coupling. Sodium hydroxide (NaOH), 2-(N-morpholino) ethane sulfonic acid (MES), N-(3-Dimethylaminopropyl)-N’-ethyl carbodiimide hydrochloride (EDC), and N-Hydroxy succinimide (NHS) were purchased from Sigma-Aldrich (St. Louis, MO, USA), and the 96-well microplate (MS-8496F) was purchased from Sumitomo Bakelite Co., Ltd. (Tokyo, Japan). Absorbance was measured by using Microplate Reader Infinite^®^F50 (Tecan Group Ltd., Mannendorf, Switzerland).

### 2.4. Evaluation of Binding Rate

The structure was rotated in a solution containing 10 ng/mL of IgA for 1~5 min, and the concentration of unreacted IgA remaining in the solution was quantified by a sandwich ELISA, using a 96-well plate. Then, the binding rate to the 3D-stack was calculated by subtracting the concentration of unreacted IgA from the initial concentration (Figure 5). The procedure used was the same as the one used in a previous study [18]. This was repeated twice, and the average value was used, which was evaluated by comparing the 3D-stack (N) and 3D-stack (C) (Table 3).

### 2.5. Conditions of Sandwich ELISA Using 3D-Stack and Conventional Method

Before utilizing 3D-stack in ELISA, a surface treatment was performed for hydrophilizing surface of the films. The 3D-stack was immersed in a 2.5 M NaOH aqueous solution as a surface treatment and hydrophilized at 50 °C for 2 h. EDC-NHS reagent was diluted 100-fold with 0.1 M MES (pH = 5.5), and then we dispensed 100 μL into the wells and then inserted a 3D-stack into the wells and rotated for 30 min. In this process, the rotation speed used was 2000 rpm. The 3D-stack was washed to remove NaOH after hydrophilization. For washing after the treatment, 100 μL of 0.05 M carbonate–bicarbonate (pH = 9.5) was dispensed into new wells, the 3D-stack was inserted, and the rotation for 20 s was repeated 5 times.

The procedure for ELISA is shown as follows. Anti-Human IgA (Bethyl Lab., Inc., Montgomery, TX, USA) was diluted 100-fold with 0.05 M carbonate–bicarbonate (pH = 9.5) into new wells, and the 3D-stack was inserted into 100 μL increments for 10 min and rotated for 10 min. After incubation, the 3D-stack was transferred to a new well, and the wash solution (50 mM Tris, 0.14 M NaCl, 0.05% Tween20, pH 8.0) was dispensed into the new well—i.e., 100 μL of each inserted into the well—and rotated for 20 s. This was repeated 5 times. For blocking, 100 μL of blocking solution (50 mM Tris, 0.14 M NaCl, 1% BSA, pH 8.0) was dispensed into new wells, 3D-stack was inserted into the wells, rotated for 10 min, and washed. After blocking, Purified Human IgA (Bethyl Lab., Inc., Montgomery, TX, USA) was diluted with 50 mM Tris, 0.14 M NaCl, 1% BSA, and 0.05% Tween20, pH 8.0, and we dispensed 100 μL into new wells; and 3D-stack was inserted into the wells, rotated for 1 min, and washed. Next, HRP Anti-Human IgA (Bethyl Lab., Inc., USA) was diluted with 50 mM Tris, 0.14 M NaCl, 1% BSA, and 0.05% Tween20, pH 8.0, and we dispensed 100 μL into new wells; 3D-stack was inserted into the wells, rotated for 1 min, and washed. Finally, TMB One-Component HRP Microwell Substrate (Bethyl Lab., Inc., USA) was dispensed in the new well in 100 μL increments, and then a 3D-stack was inserted into the wells and rotated in a darkroom for 1 min. Thereafter, the 3D-stack inserted into the well with the TMB solution was removed, and 100 μL of Stop Solution (0.12 N HCl, Sera Care., Inc., Milford, MA, USA) was dispensed into the wells. The well was inserted into a plate reader (Tecan Group Ltd., Mannendorf, Switzerland) and was measured at a wavelength of 450 nm and a reference wavelength of 620 nm. The reagent used was based on a previous study. In processes of ELISA, the rotation speed used was also 2000 rpm, but for the unsteady rotation, the condition for the transient rotation is 1 s rotation, 0.1 s stop, and acceleration and deceleration at 0.1 s each, as proposed in the previous study [18]. For conventional ELISAs using only 96-well plates, we used the protocol provided by the manufacturer (Human IgA ELISA Quantitation Set, catalog no. E80-102, Bethyl Laboratories, Inc.). The incubation time was 1 h for the capture antibody, 30 min for blocking, 1 h for the sample, 1 h for the detection antibody, and 15 min for TMB. Figure 6 shows the ELISA using the 3D-stack and the ELISA using the plate in comparison with the conventional ELISA using only a well.

## 3. Results and Discussions

### 3.1. Numerical Simulation

In evaluating the analytical model, the two planes shown in Figure 7 were extracted. First, to evaluate the flow between the films, the streamlines between the films in the XY plane are shown in Figure 8. Here, the results for 20 μm and 100 μm are shown as representative results. At 20 μm, the flow due to the centrifugal force is dominant, and at 100 μm, the flow due to the Coriolis force and the flow due to the centrifugal force are combined. Table 4 shows the change in Reynolds number when the representative velocity is set to the average flow velocity (volume average) between films, the representative length is the lamination interval of the film, and the kinematic viscosity coefficient is 1.004 × 10^−6^ m^2^/s of water. The Reynolds number changes with respect to the stacking interval, and the inertial force becomes domination around 60 μm. As a result, the circumferential flow also increased caused through inertia.

To evaluate the biomolecule transport state and the surface reaction state between films, the time change in the antigen–antibody complex concentration for the reaction time of 5 min is shown in Figure 9. From Figure 9, it was confirmed that the composite concentration increased by increasing the stacking interval, and that the disadvantage of increasing the diffusion distance was not dominant in the range of 100 μm. This is due to the concentration diffusion layer formed by the reaction. The thickness of the concentration diffusion layer varies depending on the reaction field area, flow velocity, and flow path width [29,30]. Figure 10 shows that the concentration diffusion layer of the analyte concentration formed by the surface reaction was extracted from the two planes shown in Figure 7. In Figure 10, the analyte concentration was divided by the initial concentration, and the height of the XY plane was 10 µm from the film surface. The film interval of 20 μm has a larger area with a lower concentration than other stacking intervals. In addition, it was confirmed that the radial flow was inhibited, and the area effective for the reaction was reduced because the central axis was closed by a metal pin in the notch shape. For the use of the 3D-stack, a stacking interval and rotation speed to make Re greater than 1 could be necessary for the formation of circumferential flow, and it is necessary to secure an inlet that does not impede the radial flow and circumferential flow. As a result, it is effective to secure the inlet by providing a notch.

### 3.2. Effect of the Notched Shape on Binding Rate

Figure 11 shows the variation in the binding rate per unit area for various conditions of lamination intervals and rotation times. The binding rate increases according to the rotation time in the case of 3D-stacks with an interval of 20 μm, while the binding rate remains almost saturated for the rotation time more than 1 min in the case of 3D-stacks with an interval of 100 μm. Although the tendency of the binding rate is similar for both the notched shape and annular shape, the binding rate for the notched shape is lower than that for the annular shape. The main reason is that there is flow near the central axis, and a larger dead volume that does not contribute to protein binding exists in the case of the notch shape. However, the rate of increase in the binding rate of the lamination interval and rotation conditions tended to be higher in the notch shape, as shown in Figure 12, especially for the case of a rotation time of 1 min. This confirms the formation and use of circumferential flow, and the difference in the increase rate with respect to the lamination interval indicates the importance of flow in the centrifugal direction.

### 3.3. Effect of the Notched Shape on ELISA

Based on results of the binding rate, which almost reached saturation by unsteady rotation of 1 min using the 3D-stack with film interval of 100 μm, we decided to shorten the time for each step of the ELISA test to 1 min. Therefore, in the ELISA process shown in Figure 6, the preparation process was under the same conditions as before, and only the time, n, for each process in the assay process was set to 1 min. Figure 13 shows the absorbance of ELISA at each concentration when the 3D-stack (N) of the lamination interval is 100 μm in comparison with that using a 96-well plate. In this case, the number of laminates is three for the 3D-stack (N). The coefficients of variation in the absorbance result for each condition of ELISA are shown in Figure 14. The coefficients of variation for 3D-stack are lower than 10% and similar to those of the 96-well plate. In all concentrations, the absorbance of the plate was higher than that of the 3D-stack (N), and the difference was larger at the higher concentration, but the difference was reduced at s lower concentration, and the blank value of the 3D-stack (N) was almost the same as that of the 96-well plate. This is because, at low concentrations, the concentration gradient is small, and the binding rate due to diffusion decreases in the case of the 96-well plate, but in the 3D-stack (N), the thickness of the concentration diffusion layer formed by the reaction is reduced by convection promotion, and the diffusion promotion by transient rotation is increased. However, at higher concentrations, the concentration gradient was larger, and diffusion became dominant, and the amount of binding in the 96-well plate, which took sufficient time for the reaction, was high, and the amount of binding in the 3D-stack (N) with a short reaction time was reduced. The point to emphasize is that, for 96 wells, the reaction time for each process is 1 h, while for the 3D-stack (N), it is 1 min. In other words, even though the time required for ELISA using a 3D-stack is reduced to 1/60, the reaction sensitivity is almost the same as that for 96 wells at low concentrations. This result is sufficient for practical use.

In this experiment, the number of films in the 3D-stack was set to three; it is possible to increase the number of antibodies that can be immobilized on the film surface and increase the amount of antigen adsorption by increasing the number of films. In addition, by changing the conditions for unsteady rotation and increasing the ratio of stopping time to rotation time, it is possible to further promote diffusion and solve the problem of reduction in effective surface area due to dead volume around the rotation axis. Our future work will focus on optimizing the conditions of the unsteady rotation.

## 4. Conclusions

In this study, when focusing on geometry as a convection promotion of 3D-stack, we examined the use of a cutout like a propeller to the film shape, which forms an axial flow at the notch and shortens the distance of the flow per unit time to circulation to the intervals of the film. Furthermore, by adding a notch, it is assumed that the circumferential flow from the notch will be used as an inflow between the films to improve the circulation condition in the container and between the films. The following findings were obtained by focusing on the axial flow and circumferential flow at the notch that occurs when a cutout is applied to the disc film as a shape that promotes convection, and the effect of convection promotion by the notch shape on biomolecular transport.

(1)By adding a notch to the disk, the formation of axial flow and the inflow into the film using the circumferential flow were confirmed, and it was expected that circulation was promoted by the notch.(2)It was confirmed that the Reynolds number was changed by changing the lamination spacing between the notched films, which affected the circumferential flow in the film, and that convection promotion reduced the concentration diffusion layer and promoted the antigen–antibody reaction.(3)From the coupling rate evaluation, the formation and utilization of the circumferential flow were confirmed, and the coupling speed of the 3D-stack was increased regardless of the notch shape for the lamination spacing.(4)In the sensitivity evaluation, the sensitivity of the plate was higher, but it was confirmed that the difference in the binding amount was narrowed in the low-concentration region (6.25 and 3.125 ng/mL), indicating that the circulation promotion by the notch shape was effective for both the low concentration and rapid concentration.

## Figures and Tables

**Figure 1 micromachines-15-00613-f001:**
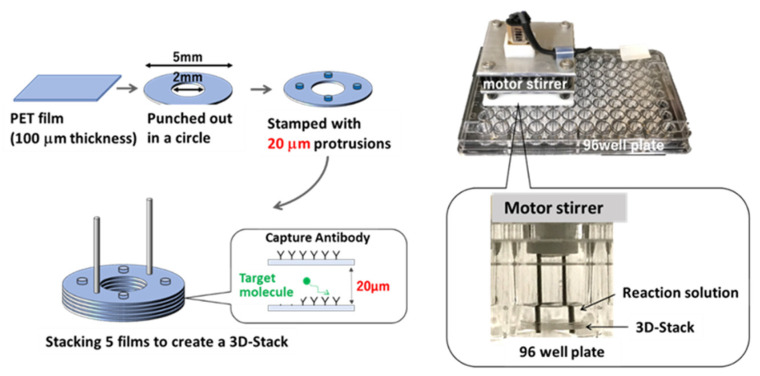
Schematic image for fabrication of 3D-stack and utilization within a well.

**Figure 2 micromachines-15-00613-f002:**
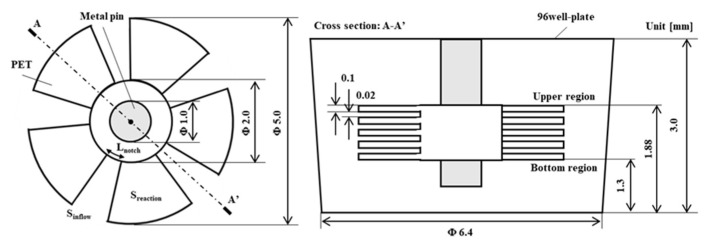
Schematic illustrations of notch structure (**left**, plan view; **right**, front view).

**Figure 3 micromachines-15-00613-f003:**
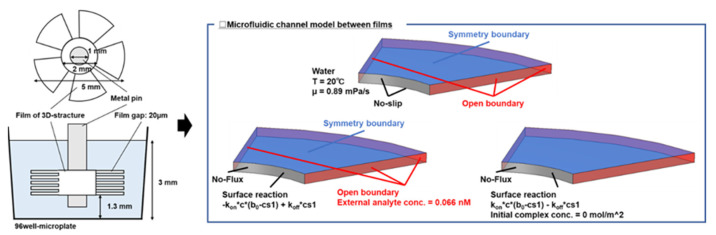
Schematic diagram of the analysis model.

**Figure 4 micromachines-15-00613-f004:**
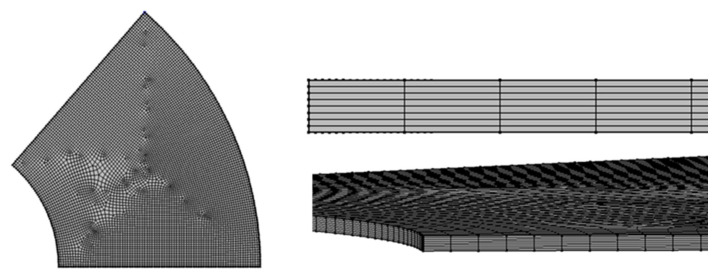
Mesh model for FE analysis.

**Figure 5 micromachines-15-00613-f005:**
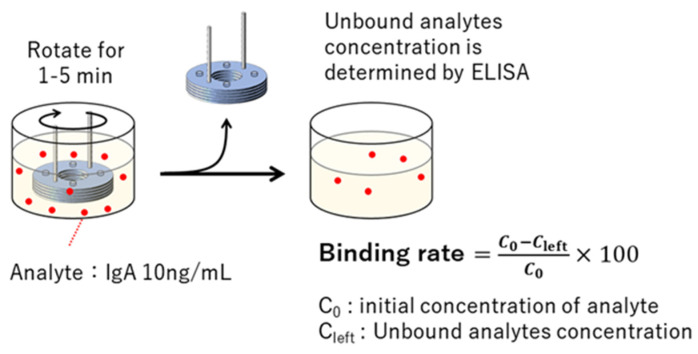
Procedure for coupling ratio evaluation [18].

**Figure 6 micromachines-15-00613-f006:**
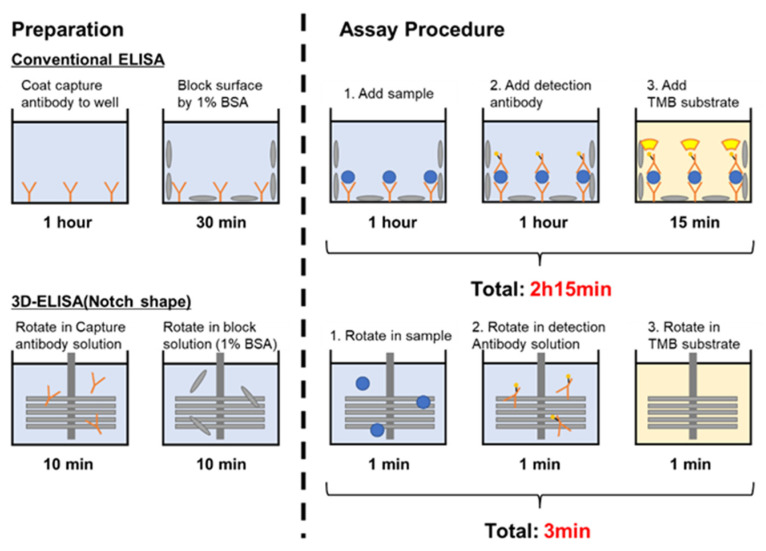
ELISA procedures for using 3D-stack and conventional well.

**Figure 7 micromachines-15-00613-f007:**
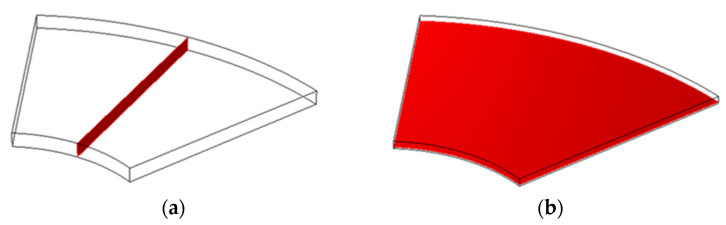
Evaluated cross-section: (**a**) y-z plane and (**b**) x-y plane.

**Figure 8 micromachines-15-00613-f008:**
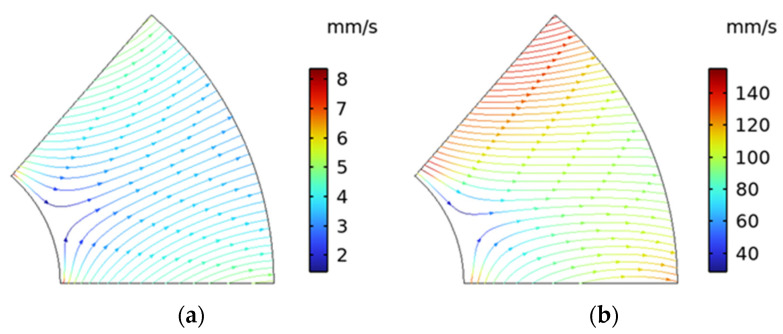
Streamline and velocity color map: (**a**) 20 µm and (**b**) 100 µm.

**Figure 9 micromachines-15-00613-f009:**
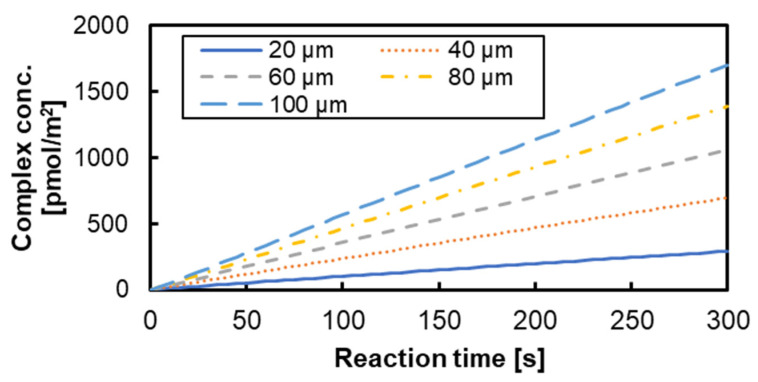
Variation in antigen and antibody compound-combined body concentration.

**Figure 10 micromachines-15-00613-f010:**
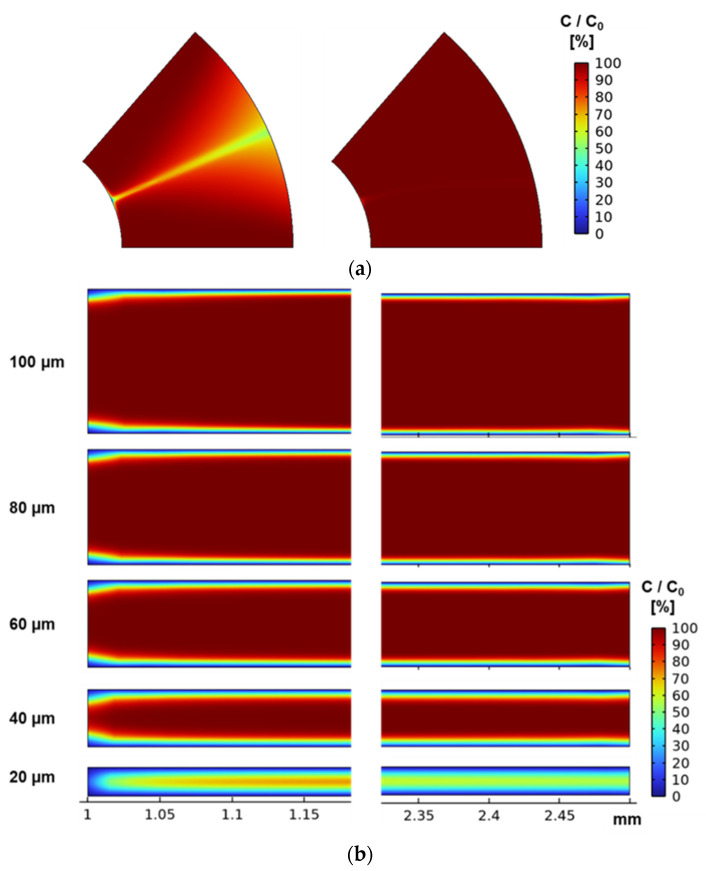
Analyte concentration distribution between films in each plane. (**a**) Analyte concentration distribution in the XY plane (left: 20 μm, right: 100 μm). (**b**) Analyte concentration distribution in the YZ plane.

**Figure 11 micromachines-15-00613-f011:**
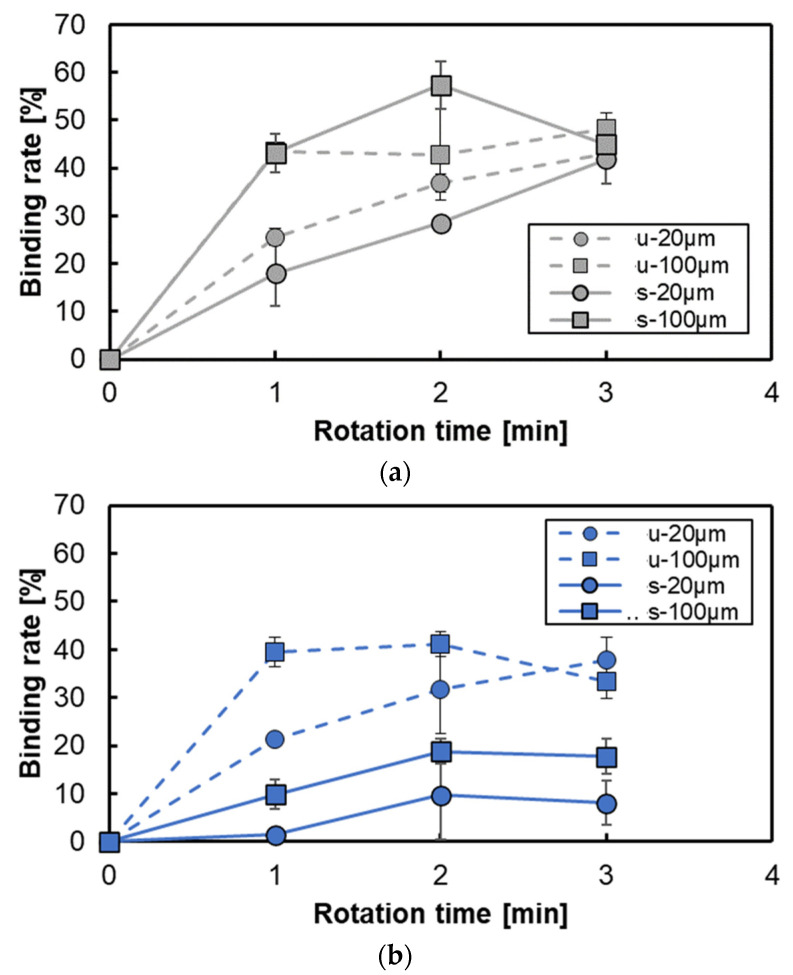
Variation in the binding rate for each shape: (**a**) for annular shape with intervals of 20 μm and 100 μm, and (**b**) for notched shape with intervals of 20 μm and 100 μm. u, unsteady rotation; s, steady rotation.

**Figure 12 micromachines-15-00613-f012:**
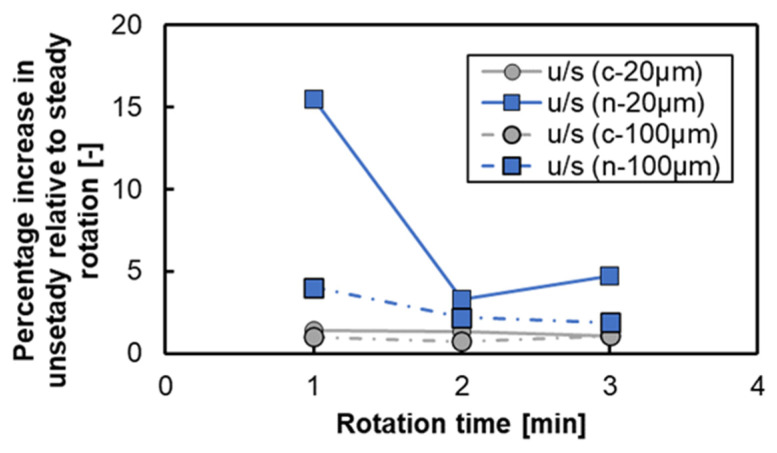
Binding ratio of unsteady to steady rotation. u/s, unsteady/steady; c, annular shape; n, notched shape.

**Figure 13 micromachines-15-00613-f013:**
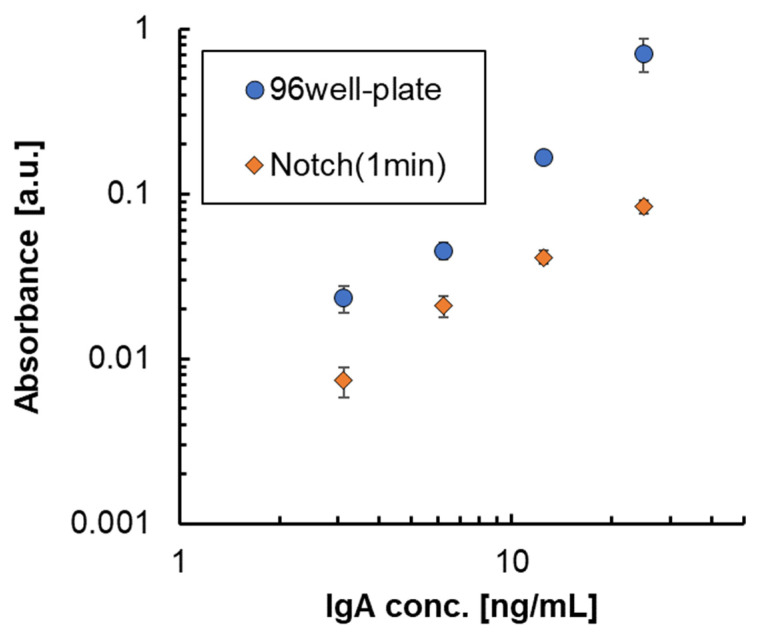
Absorbance of ELISA at each concentration.

**Figure 14 micromachines-15-00613-f014:**
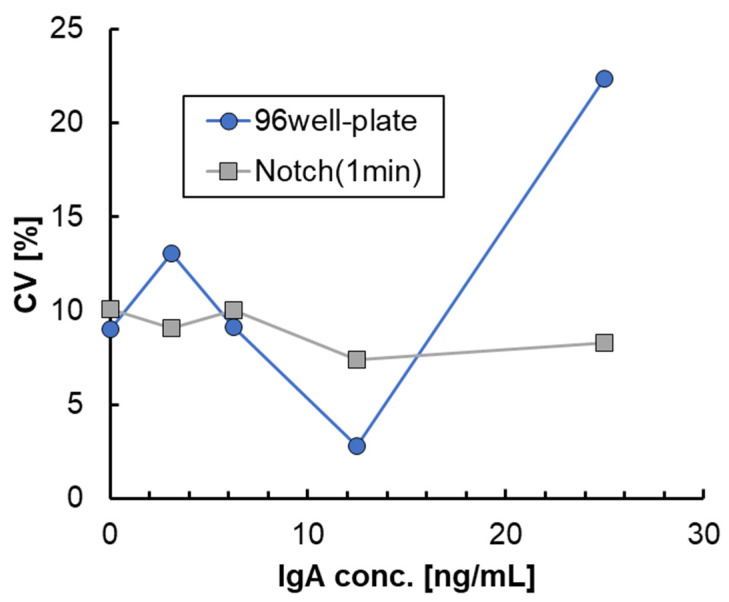
Coefficient of variation in the result of absorbance for each condition of ELISA.

**Table 1 micromachines-15-00613-t001:** Numerical values of the parameters used in the analysis.

Parameter	Value
Rotation speed	2000 rpm
Viscosity (water)	0.89 mPa·s
Density (water)	997 kg/m^3^
Association constant (*k_on_*)	1 × 10^5^ m^3^/mol/s
Dissociation constant (*k_off_*)	1 × 10^−4^ 1/s
Surface diffusion coefficient (*D_s_*)	1 × 10^−9^ m^2^/s
Diffusion coefficient (*D*)	4 × 10^−7^ cm^2^/s
Immobilized ligand concentration (*B*_0_)	1 × 10^−11^ mol/cm^2^

**Table 2 micromachines-15-00613-t002:** Number of elements for each geometry.

	Element Number
Film Gap: 20, 40, 60, 80, 100 (µm)	68.3, 136, 205, 273, 341 (×10^4^)

**Table 3 micromachines-15-00613-t003:** Sample conditions for coupling ratio.

	Type	Film Gap	
		20 µm	100 µm
3D-stack	3D-stack (N)	3D-stack(N)-20	3D-stack(N)-100
	3D-stack (C)	3D-stack(C)-20	3D-stack(C)-100

**Table 4 micromachines-15-00613-t004:** Change in mean flow velocity and Reynolds number at each film interval.

Film Gap (mm)	Flow Velocity (mm/s)	*Re*
0.02	2.45	0.0485
0.04	9.82	0.389
0.06	22.2	1.32
0.08	40.4	3.19
0.1	65.7	6.51

## Data Availability

The data presented in this study are available upon request from the corresponding author.

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
