# Peer review of "Enhancement of Convection and Molecular Transport into Film Stacked Structures by Introduction of Notch Shape for Micro-Immunoassay"

_micromachines, 2024, doi:10.3390/mi15050613_

Round 1

Reviewer 1 Report

Comments and Suggestions for Authors

The authors developed a 3D-Stack microfluidic device that can be used in combination with 96-well plates for 10 micro-immunoassay. And the performance of it was analyzed. Here are some questions that need to be revised or addressed before publication.

1.      Page 5, equation 6 needs to be confirmed the correction.

2.      Page11-12, fig 11-12, which is the meaning of c/u, u/s, n/s, n/u and u/s?

3.      Page 13, fig 13, based on the result, the sensitivity of ELISA is higher than 3D-stack.

4.      Some of the references are too old, it would be good to change them to new ones. The results of the standard curves need to be discussed and the standard-deviations need to be added.

Comments on the Quality of English Language

Minor editing of English language required.

Reviewer 2 Report

Comments and Suggestions for Authors

The manuscript by Arau et al, presents an enhancement of the convection and the molecular transport into film stacked structures, an Elisa based assay developed by the group,  by introduction of Notch shape

The manuscript is well organizde, the shcemes let to a good understanding of the manuscript and the figures are well described.

Couple of small errors are present in the text:

Line 149 and 164 – an error occurred with the reference manager and is present in the text

Line 190 “To evaluate the independence of the mesh to the mesh,”

Reviewer 3 Report

Comments and Suggestions for Authors

Introduction:

Line 24: The authors should consider citing a scientific literature source rather than a government website as Reference 1. Consider Baker et al, Nat Rev Microbiol 20, 193-205 (2022), doi:10.1038/s41579-021-00639-z

Line 34: If this is a quotation, the authors must cite the source that they are quoting.

Line 58: Molds for soft lithography do not need to be produced using photolithography, other techniques such as 3D printing are available.

Line 61: It is not immediately obvious to the reader that the device described here is in fact a microfluidic device. While it could be argued that the device relies for its function on the behavior of fluid at low Reynolds number, this claim is not made explicitly anywhere in the paper. While this paper claims that the similar device described by the authors in ref.16 is a microfluidic device, ref. 16 does not say that it is one.

Given the important differences between this device and more classical channel-based microfluidic devices, the authors need to either make their reasoning for why they describe it as a microfluidic device clearer or remove references to its being one.

Line 64: It appears from here (and from Figure 5 later in the paper) that the ELISA reaction takes place on the 3D-Stack device not on the surface of the well plate. Is this correct? If so, this needs to be made clearer.

Line 67: The authors describe the device as “easy to operate when automated”. Is it possible to clarify how it fits into existing automation protocols?

Line 67: While the reaction times are longer for conventional well-plate ELISA than for the 3D-Stack system, well-plate ELISA has the advantage of much easier parallelization, particularly with the use of tools such as multichannel/repeat pipettors or liquid-handling robots. Therefore, total time (and total ‘hands-on’ time) for an experiment incorporating multiple replicates or a range of different conditions may not actually be decreased. The authors need to address this issue. One option would be to compare the time taken for experiments with multiple samples- another would be to describe a way of parallelizing the 3D-Stack method.

Line 68: It needs to be made clearer that this study used the 3D-stack method.

Line 88: Is the (1) supposed to be a reference? If so, the reference is missing. Also, the meaning of “meridional reflux” is not necessarily clear in English- a diagram illustrating the desired and undesired flow paths would be useful here.

Line 90: Ref. 19 links to a lab introductory page in Japanese. Given that the Tesla pump has been known for a long time, and that this page is not original research, this should be replaced with an English-language reference- either Nikola Tesla’s original patent (US1061206A) or a scientific paper.

Methods:

Line 122: How are the device parts cut and assembled? If they are cut by hand, this needs to be stated- if a laser cutter, plotter cutter (xurography), etc is used then the model should be named. Similarly, if an adhesive is used for assembly readers need to know which one, while if assembly is by hot pressing without adhesive the parameters should be stated.

Lines 149-50, 164, 182-184 Missing references!

Line 212: The source of the PET film should be detailed in the section describing the device assembly.

Line 217: Was the 3D-stack washed at all to remove NaOH after hydrophilization?

Line 218: During the 3D-stack based ELISA assay, what rotation speed was used? It appears from the results section that unsteady and steady rotation were compared, but the design of this experiment is not clear in the Methods section.

Line 256: The design of the coupling ratio evaluation experiment is not clear.

 Results:

In general, it is not clear which part of the Methods section corresponds to which part of the Results section.

Line 297: Please clarify why a Reynolds number greater than 1 is necessary. Normally any change of flow regime only occurs at Re significantly higher than 1.

Line 351 and figure 13: Figure 13 does not have error bars. Therefore, it is difficult to evaluate the performance of the ELISA using a 3D-Stack. It appears that the 3D-Stack ELISA works as a qualitative test to determine the presence/absence of the analyte, but may be less suited to quantitative measurements of analyte concentration. Is this correct? If so, this should be described in the text of the paper.

(If the error in absorbance is small enough that it is obscured by the markers in the figure, different markers should be used in order to make this clear)

Lines 354-7: The authors claim that the problem of reduced effective surface area can be solved by changing the conditions of “unsteady” rotation. If this has been done successfully, it should be detailed in the body of the paper.  If not, they should make it clear that this is only a theoretical possibility so far.

Discussion:

Line 371: The word “expected” suggests a hypothesis. Do the authors mean to say that the hypothesis was shown to be correct, or that they suspect this but cannot show it?

References:

References

Ref 2 needs a DOI instead of the partial/incorrect URL

References 7-12 need DOIs

Ref. 16 does not have a DOI, and the issue/article number is incorrect. This needs to be corrected.

References 24-26 are incomplete. All need journal titles and DOIs. In addition, References 25 and 26 should list all the authors rather than only one.

Comments on the Quality of English Language

Line 36: “the tests” should be “tests”

Line 39-40: “Enzyme-linked… assay”- Meaning of this sentence is unclear, rephrase

Line 47: “It is expected” should be “the use of microfluidics is expected”

Line 53: “the assays” should be “these assays”

Lines 96-102: These sentences are difficult to understand, please rephrase.

Line 143: Check spelling of Navier-Stokes

Line 190: “To evaluate the independence of the mesh to the mesh”- check this sentence.

Line 265: “Here, 20,100 μm is represented by 20,100 μm- check this sentence.

Line 271: “Domination” should be “dominant”

Line 272-275: Part of the article template has been left in.

Line 282: There are two tables numbered Table 3. The second of these needs to be renumbered.

Line 308: This title should be changed, currently it looks like the authors are trying different shapes of notch.

Line 311, 331: μ is missing

Line 318: Sentence fragment (beginning “Especially”)

Discussion

Line 363: Meaning of “shortens the distance of the flow per unit time” is unclear. Is the velocity reduced?

Line 370: The notch was added, not attached- “attaching” something cannot describe subtracting material, as here.

Round 2

Reviewer 3 Report

Comments and Suggestions for Authors

The authors are thanked for their quick return of the revised manuscript. There are still a few points which need to be made, as follows:

Intro Rebuttal Point 7: The ability to drive 16 3D-Stacks simultaneously with one motor dramatically increases the utility of the 3D-stack method. The authors should mention this in the paper.

Intro Rebuttal Point 8: This part does not appear to have been changed. A suitable correction would be "In previous studies using the 3D-stack, Maeno et al..." (emphasis added to show change)

Methods Rebuttal Point 4: The fact that the 3D-stack was washed to remove NaOH after hydrophilization (and what it was washed with) needs to be added to the paper.

Results Rebuttal Point 4: It needs to be made clearer in the text that future work will focus on optimizing the conditions of "unsteady" rotation 

Comments on the Quality of English Language

It appears that the meaning of the phrase "shortens the distance of the flow per unit time" in Line 363 has not been clarified. This still needs to be done.
